# Genome insights from the identification of a novel *Pandoraea sputorum* isolate and its characteristics

**Rui-Fang Gao**[1,2]*, **Ying Wang**[1,2], **Ying Wang**[1,2], **Zhi-Wen Wang**[3], **Gui-Ming Zhang**[1,2]

**1** Animal & Plant Inspection and Quarantine Technology Center of Shenzhen Customs District P.R. China, Shenzhen, China, **2** Shenzhen Key Laboratory for Research & Development on Detection Technology of Alien Pests, Shenzhen Academy of Inspection and Quarantine, Shenzhen, China, **3** PubBio-Tech Services Corporation, Wuhan, China

* gaoruifang606@sina.com

**Data Availability Statement:** All relevant data are within the manuscript and its Supporting Information files.

## Abstract

In this study, we sequenced a bacteria isolate *Pandoraea* sp. 892iso isolated from a *Phytophthora rubi* strain which is an important plant pathogenic oomycete, identified through genome and combined the data with existing genomic data from other 28 the genus of *Pandoraea* species. Next, we conducted a comparative genomic analysis of the genome structure, evolutionary relationships, and pathogenic characteristics of *Pandoraea* species. Our results identified *Pandoraea* sp. 892iso as *Pandoraea sputorum* at both the genome and gene levels. At the genome level, we carried out phylogenetic analysis of single-copy, gene co-linearity, ANI (average nucleotide identity) and AAI (average amino acid identity) indices, *rpoB* similarity, MLSA phylogenetic analysis, and genome-to-genome distance calculator calculations to identify the relationship between *Pandoraea* sp. 892iso and *P. sputorum*. At the gene level, the quorum sensing genes *ppn*I and *ppn*R and the *OXA-159* gene were assessed. It is speculated that *Pandoraea* sp. 892iso is the endosymbiont of the Oomycetes strain of *Phytophthora rubi*.

## Introduction

The genus *Pandoraea*, originating from the term "Pandora's box", refers to the source of all evil in Greek mythology and was established by Coenye et al. in 2000 [1]. The species are characterized as nonspore-forming, catalase-positive, aerobic, gram-negative rods with polar flagella. Some species in this genus were once identified closest to *Burkholderia cepacia* complex (Bcc), *Ralstonia pickettii*, or *Ralstonia paucula* based on phenotype. [1–3]. The genus *Pandoraea* includes 28 named species (*Pandoraea anapnoica*, *P. anhela*, *P. apista*, *P. aquatica*, *P. bronchicola*, *P. capi*, *P. captiosa*, *P. cepalis*, *P.commovens*, *P. communis*, *P. eparura*, *P. faecigallinarum*, *P. fibrosis*, *P. horticolens*, *P. iniqua*, *P. morbifera*, *P. norimbergensis*, *P. nosoerga*, *P. oxalativorans*, *P. pneumonica*, *P. pnomenusa*, *P. pulmonicola*, *P. soli*, *P. sputorum*, *P. terrae*, *P. terrigena*, *P. thiooxydans* and *P. vervacti* [3–5]. *Pandoraea* sp. types have been predominantly isolated from patients with septicemia or respiratory tract infections (mostly cystic fibrosis), as well as from food, water, soil, and food [2, 4, 6–9].

**Funding:** Grant funds of National Key R&D Programme of China (No. 2016YFF0203204) and National Key Technology Research and Development Programme of China (No. 2012BAK11B06) for Gui-Ming Zhang are used in study design, data collection and analysis. The funds of Scientific Research Project of General Administration of Customs. P. R. China (No. 2021HK171) for Ying Wang is used for the decision to publish, and preparation of the manuscript.

**Competing interests:** The authors have declared that no competing interests exist.

Clinical manifestations of this terrorizing pathogen revolve around nosocomial infections and its ability to deteriorate lung function and even cause multiple organ impairment [10–12]. These organisms appear to be potential pathogens for individuals with cystic fibrosis as well for cross-infection [13]. Further, *Pandoraea* spp. isolated from environmental samples have considerable potential for biotechnological application given various beneficial degradation abilities, such as removing isomers of 1,2,3,4,5,6-hexachlorocyclohexane (HCH) [13], catalyzing the aerobic transformation of biphenyl and various polychlorinated biphenyls (PCBs) [14, 15], catalyzing the decarboxylation of 2,6-dihydroxybenzoate and regioselective carboxylation of 1,3-dihydroxybenzene to 2,6-dihydroxybenzoate, catalyzing the regioselective carboxylation of phenol and 1,2-dihydroxybenzene [16], degrading kraft lignin without any cosubstrate under high alkaline conditions [17], degrading chlorobenzene [18], biodegrading endosulfan classified as an organochlorine pesticide [19], treating malachite green [20], and metabolizing oxalate [21].

Reflecting on previous research, *Pandoraea* spp. have frequently been misidentified in many clinical laboratories, leading to a lack of clinical documentation on their virulence potential. Therefore, it is important to accurately identify *Pandoraea* spp.. Earlier classification of prokaryotes was based solely on phenotypic similarities [22], but modern prokaryote characterization has been strongly influenced by advances in genetic methods. One criterion to be considered a species is to be essentially a collection of types that are characterized by at least one diagnostic phenotypic trait and to have purified DNA molecules that show at least 70% cross-hybridization (DNA-DNA hybridization, DDH) [22–25]. This is pragmatic and universally applicable within the bacterial domain, while the lack of this standard has been increasingly found when it comes to reliable diagnosis of infectious disease agents, international regulations for transport, quarantine, and so on [26–28]. Subsequently, this parameter has been applied most frequently in species identification at the whole genome level [29–32]. Genome Blast Distance Phylogeny (GBDP) [33], the core and pangenome [32], and the genomic-distance index based on DNA maximal unique matches (MUM) [34] are used to identify new species. Unfortunately, our understanding of *Pandoraea* spp. at the genomic level is relatively superficial, whereby the majority of the literature focuses principally on the usage of genotypic data to facilitate accurate genus- and species-level identification and secondarily on biotechnological potential [1, 2, 18, 21].

In the present study, suspected bacteria isolated from an oomycete strain was identified through whole genome sequencing. The taxonomic status of this isolate was verified at the genome and gene levels, and its phylogenetic relationship with similar species was explored using indices, such as ANI/AAI, MLSA (Multi-locus Sequence Analysis) phylogenetic analysis, genome-to-genome distance calculations, quorum sensing, and oxacillinase gene analysis.

## Materials and methods

### Strains, cultures, and DNA extraction

When we performed morphological observations on the hyphae of a *Phytophthora rubi* strain (No. 109892) from Westerdijk Fungal Biodiversity Institute, we inadvertently discovered the structure of suspected bacteria present in the mycelia. The structure still existed after the isolation by monofilament isolation and monospore isolation of the fungus. After isolation and culture, we obtained an analytical strain of bacteria, so that part of the name of which is called "892iso isolate". Separation, purification, and culture were carried out on beef extract peptone medium plates at 30°C for 48 h. A TIANamp Bacteria DNA Kit (Tiangen, China) was used for genomic DNA.

## Sequencing, assembly, and annotation

The whole genome was sequenced and assembled by a strategy that combined paired-end and mate-paired libraries. One targeted insert size of 500 bp was constructed using the TruSeq Nano DNA LT Library Prep Kit (Illumina, USA). One mate-paired library (2 kb) was constructed by the Nextera Mate Pair Sample Prep Kit (Illumina, FC-132–1001, USA) on the Illumina HiSeq 2500 platform. SOAPdenovo (v2.04) was used for *de novo* assembly. The assembled genome was annotated with a web-based tool called RAST (http://rast.nmpdr.org). RAST can identify repeat sequences in the genome, protein-encoding rRNA and tRNA genes, and assign functions to the genes.

## Whole genome alignment and some indices calculation

Mauve (version 2.3.1) was used to align genomes for synteny analysis. The calculation of ANI and AAI was based on BLAST alignment results using a Perl script. The genome-to-genome distance calculator calculations were based on a web server (https://ggdc.dsmz.de/) that uses multi-FASTA files as input. The *ppnI*/*ppnR* genes of *P. pnomenusa* were download from NCBI (accession ID KF887500.1 and KF900148.1), then aligned with all *Pandoraea* gene sets, all matches with the identity greater than 0.3 and score greater than 100 were retained. The *ppnI* candidates should contain PF00765 domain and *ppnR* candidates contain PF03472 domain, and the candidate pairs should be adjacent to each other. An intrinsic Carbapenem-Hydrolyzing Oxacillinases gene of *Pandoraea* sp. HD7676 was download from NCBI (accession ID: KP771987.1). BLAST was employed to identify homolog genes in the 28 *Pandoraea* species.

## Comparative genome analysis

All protein sequences in reference genomes were downloaded and set as the query for all-vs-all BLASTP. OrthoMCL (version 2.0.8) was used to identify single-copy genes with I (inflation) set at 1.5. Next, MUSCLE (version 3.8.425) was used to align the sequences of the associated proteins. PAL2NAL (version 14.0) was used to convert the protein alignment to codon alignment. Gblock (version 0.91b) was used to remove the alignment results that were deemed unreliable. The phylogenetic tree was built by single-copy genes, with *Burkholderia cepacia* strain LO6 as the outgroup. MCMCTree software in PAML (version 4.7) was used to estimate the divergence time. CAFÉ (version 4) was used to calculate the expansion and contraction of these gene families.

# Results

## Genome assembly, annotation, and validation of protein-coding genes

The genome of the *Pandoraea* sp. 892iso isolate was assembled from sequencing data generated by HiSeq 2000 by SOAPDenovo2 assembler. The total length of the top 48 longest scaffolds was 5.83 Mb, representing approximately 82.6-fold genome sequence coverage. The N50 and maximum lengths of scaffolds was 1.43 kb. Most of the length was concentrated on 12 scaffold sequences over 1,000 bp, of which the longest sequence was 2.06 MB (Fig 1). A total of 5,367 protein-coding genes were predicted from the genome assembly, 5,131 (95.60%) of which were supported by the RNA-seq data (coverage > = 90%). Within these protein-coding genes, 4,274 (79.63%) were assigned a biological function. Among the 1,093 ORFs without known function, 736 showed similarity to other database entries. For *Pandoraea* sp. 892iso isolate, the coding regions from the predicted genes constituted 88.61% of the genome (total length of all genes divide the genome size) and the average gene density was

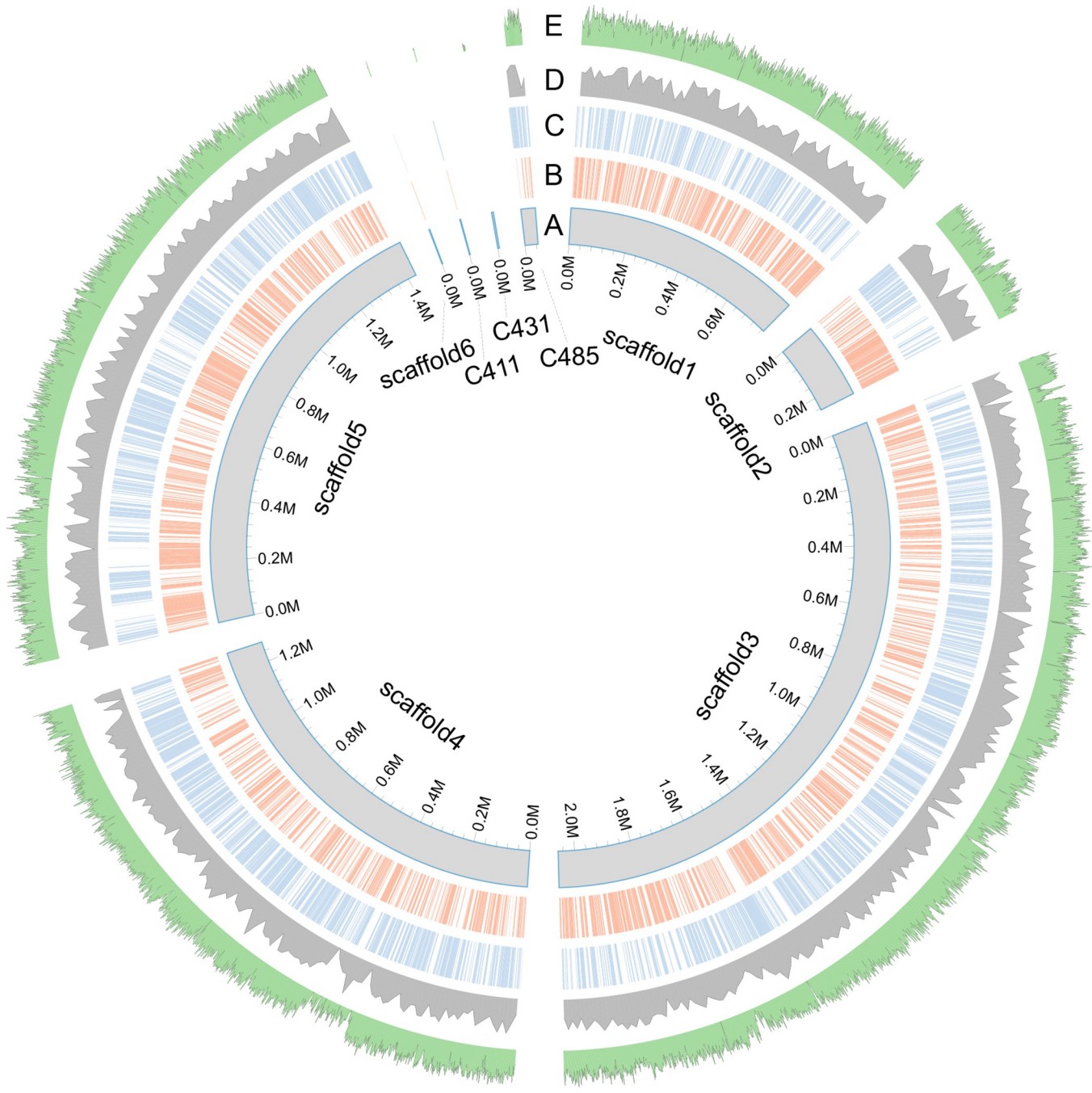

**Fig 1. Structure of the genome assembly.**

919 genes per 1 Mb (total number of all genes divide the genome size, times with 100000bp), which were more or fewer than most of other sequenced *Pandoraea* species. The GC content of the genome, coding sequences, and repetitive elements were 62.66%, 63.32%, and 57.52%, respectively. A total of 63 tRNA genes were predicted from the assembly. The genome characteristics of *Pandoraea* sp. 892iso and other *Pandoraea* species are shown in Table 1.

**Table 1. Genome and gene comparison of *Pandoraea* sp. 892iso and other *Pandoraea* species.**

| Content | | *Pandoraea* sp. 892iso | *Pandoraea anapnoica* | *Pandoraea apista* | *Pandoraea aquatica* | *Pandoraea bronchicola* | *Pandoraea capi* | *Pandoraea captiosa* | *Pandoraea cepalis* | *Pandoraea commovens* | *Pandoraea communis* |
|---|---|---|---|---|---|---|---|---|---|---|---|
| Accession number | | GCF_902459725.1 | GCF_902459765.1 | GCF_001465595.2 | GCF_902459565.1 | GCF_902459805.1 | GCF_902459735.1 | GCF_902459775.1 | GCF_902459625.1 | GCF_902459615.1 | GCF_902459745.1 |
| Genome | Scaffold Number | 48 | 48 | 2 | 17 | 34 | 31 | 36 | 32 | 26 | 17 |
| | Total Length (Mb) | 5.83 | 6.13 | 5.57 | 5.96 | 5.35 | 5.85 | 6.14 | 5.16 | 6.04 | 5.71 |
| | GC Content (%) | 62.7 | 62.4 | 62.63 | 62.89 | 62.97 | 63.44 | 63.3 | 63.54 | 62.63 | 62.57 |
| | N50 Length | 1,430,084 | 278,466 | - | 442,715 | 323,330 | 401,082 | 280,257 | 286,967 | 434,611 | 434,208 |
| | N90 Length | 768,040 | 128,458 | - | 244,118 | 147,078 | 151,211 | 137,159 | 88,384 | 144,451 | 240,155 |
| | Longest scaffold | 2,057,907 | 677,688 | - | 1,296,496 | 977,860 | 812,744 | 1,307,560 | 1,062,234 | 1,441,918 | 1,422,275 |
| Gene | Gene Number | 5367 | 5,348 | 4,969 | 5,197 | 4,734 | 5,049 | 5,328 | 4,602 | 5,246 | 5,051 |
| | Gene Length (bp) | 5,175,057 | 5,334,275 | 4,830,549 | 5,211,988 | 4,656,675 | 5,074,754 | 5,326,122 | 4,478,447 | 5,288,517 | 4,979,502 |
| | GC Content in Gene Region (%) | 63.32 | 63.11 | 63.21 | 63.6 | 63.52 | 64.13 | 63.89 | 64.03 | 63.33 | 63.13 |
| | Gene Length/Genome (%) | 88.61 | 87.07 | 86.7 | 87.48 | 87.02 | 86.72 | 86.75 | 86.8 | 87.6 | 87.23 |
| | Gene Average Length (bp) | 964 | 997 | 972 | 1,003 | 984 | 1,005 | 1,000 | 973 | 1,008 | 986 |
| | Intergenic Region Length (bp) | 665,083 | 792,413 | 740,711 | 746,139 | 694,448 | 777,390 | 813,460 | 681,119 | 748,432 | 729,101 |
| | GC Content in Intergenic Region (%) | 57.52 | 57.63 | 58.9 | 57.95 | 59.27 | 58.97 | 59.45 | 60.3 | 57.73 | 58.74 |
| | Intergenic Region Length/Genome (%) | 11.39 | 12.93 | 13.3 | 12.52 | 12.98 | 13.28 | 13.25 | 13.2 | 12.4 | 12.77 |

| Content | | *Pandoraea eparura* | *Pandoraea faecigallinarum* | *Pandoraea fibrosis* | *Pandoraea horticolens* | *Pandoraea iniqua* | *Pandoraea morbifera* | *Pandoraea norimbergensis* | *Pandoraea nosoerga* | *Pandoraea oxalativorans* | *Pandoraea pneumonica* | *Pandoraea pmomenusa* |
|---|---|---|---|---|---|---|---|---|---|---|---|---|
| Accession number | | GCF_902459725.1 | GCF_001029105.3 | GCF_000807775.2 | GCF_902459555.1 | GCF_902459685.1 | GCF_902459575.1 | GCF_001465545.3 | GCF_902459585.1 | GCF_000972785.3 | GCF_902459645.1 | GCF_000504585.2 |
| Genome | Scaffold Number | 35 | 3 | 1 | 68 | 17 | 47 | 1 | 41 | 5 | 12 | 1 |
| | Total Length (Mb) | 5.21 | 5.73 | 5.59 | 6.01 | 6.34 | 5.23 | 6.17 | 4.86 | 6.5 | 5.85 | 5.39 |
| | GC Content (%) | 63.68 | 63.45 | 62.82 | 62.31 | 63.06 | 64.65 | 63.06 | 66.13 | 63.08 | 62.45 | 64.89 |
| | N50 Length | 259,402 | - | - | 290,798 | 382,973 | 316,192 | - | 229,370 | - | 265,947 | - |
| | N90 Length | 102,841 | - | - | 73,897 | 241,289 | 80,719 | - | 91,075 | - | 5,636 | - |
| | Longest scaffold | 893,217 | - | - | 787,753 | 1,308,188 | 801,833 | - | 664,052 | - | 2,096,772 | - |
| Gene | Gene Number | 4,615 | 5,027 | 4,855 | 5,322 | 5,499 | 4,652 | 5,356 | 4,297 | 5,648 | 5,168 | 4,759 |
| | Gene Length (bp) | 4,496,889 | 4,932,939 | 4,868,583 | 5,167,287 | 5,558,312 | 4,536,412 | 5,418,712 | 4,198,421 | 5,522,745 | 5,131,811 | 4,684,824 |
| | GC Content in Gene Region (%) | 64.17 | 63.98 | 63.39 | 62.98 | 63.76 | 65.1 | 63.72 | 66.51 | 63.59 | 63.06 | 65.36 |
| | Gene Length/Genome (%) | 86.39 | 86.05 | 87.06 | 86 | 87.68 | 86.68 | 87.86 | 86.35 | 84.96 | 87.8 | 86.98 |

(*Continued*)

**Table 1.** (Continued)

| Content | Pandoraea sp. 892iso | Pandoraea anapnoica | Pandoraea anhela | Pandoraea apista | Pandoraea aquatica | Pandoraea bronchicola | Pandoraea capi | Pandoraea captiosa | Pandoraea cepalis | Pandoraea commovens | Pandoraea communis |
|---|---|---|---|---|---|---|---|---|---|---|---|
| Gene Average Length (bp) | 974 | 981 | 1,003 | 971 | 1,011 | 975 | 1,012 | 977 | 978 | 993 | 984 |
| Intergenic Region Length (bp) | 708,688 | 799,725 | 723,482 | 841,203 | 780,817 | 696,886 | 748,658 | 663,693 | 977,986 | 713,267 | 701,122 |
| GC Content in Intergenic Region (%) | 60.55 | 60.15 | 58.94 | 58.19 | 58.13 | 61.77 | 58.29 | 63.71 | 60.23 | 58.09 | 61.71 |
| Intergenic Region Length/Genome (%) | 13.61 | 13.95 | 12.94 | 14 | 12.32 | 13.32 | 12.14 | 13.65 | 15.04 | 12.2 | 13.02 |

| Content | Pandoraea pulmonicola | Pandoraea soli | Pandoraea sp. XY-2 | Pandoraea sputorum | Pandoraea terrae | Pandoraea thiooxydans | Pandoraea vervacti |
|---|---|---|---|---|---|---|---|
| Accession number | GCF_000815105.2 | GCF_902459595.1 | GCF_004193915.1 | GCF_900187205.1 | GCF_902459695.1 | GCF_001017775.3 | GCF_000934605.2 |
| **Genome** | | | | | | | |
| Scaffold Number | 1 | 51 | 1 | 1 | 81 | 1 | 2 |
| Total Length (Mb) | 5.87 | 4.96 | 5.06 | 5.74 | 6.18 | 4.46 | 5.74 |
| GC Content (%) | 64.3 | 63.62 | 63.76 | 62.78 | 62.79 | 63.19 | 63.52 |
| N50 Length | - | 370,563 | - | - | 194,136 | - | - |
| N90 Length | - | 61,129 | - | - | 60,237 | - | - |
| Longest scaffold | - | 921,398 | - | - | 456,896 | - | - |
| **Gene** | | | | | | | |
| Gene Number | 4,996 | 4,393 | 4,512 | 4,994 | 5,590 | 4,091 | 4,889 |
| Gene Length (bp) | 5,040,965 | 4,324,589 | 4,386,412 | 5,002,422 | 5,421,742 | 3,998,582 | 4,955,787 |
| GC Content in Gene Region (%) | 65 | 64.13 | 64.26 | 63.5 | 63.31 | 63.76 | 64.11 |
| Gene Length/Genome (%) | 85.91 | 87.15 | 86.75 | 87.1 | 87.78 | 89.57 | 86.39 |
| Gene Average Length (bp) | 1,009 | 984 | 972 | 1,002 | 970 | 977 | 1,014 |
| Intergenic Region Length (bp) | 826,656 | 637,393 | 669,794 | 740,701 | 755,081 | 465,604 | 780,495 |
| GC Content in Intergenic Region (%) | 59.97 | 60.12 | 60.47 | 57.89 | 58.98 | 58.3 | 59.82 |
| Intergenic Region Length/Genome (%) | 14.09 | 12.85 | 13.25 | 12.9 | 12.22 | 10.43 | 13.61 |

## Comparative genomics and identification at the genome level

**Comparative genomic analysis.** A total of genes in *Pandoraea* sp. 892iso were classified through cluster analysis. The distribution of best hits within the genus *Pandoraea* is shown in Fig 2. In total, 3,849 orthologous genes were shared in common between *Pandoraea* sp. 892iso and the other four *Pandoraea* species. The cluster analysis of *Pandoraea* sp. 892iso, 28 *Pandoraea* species, and *Burkholderia cepacia* as an outgroup was carried out by orthoMCL to obtain the result of a common single-copy gene family. The phylogenetic relationship in view of these single-copy genes is shown in Fig 3 and S1 Table, which shows the closest phylogenetic relationship to be between *Pandoraea* sp. 892iso and the *P. sputorum* strain DSM21091.

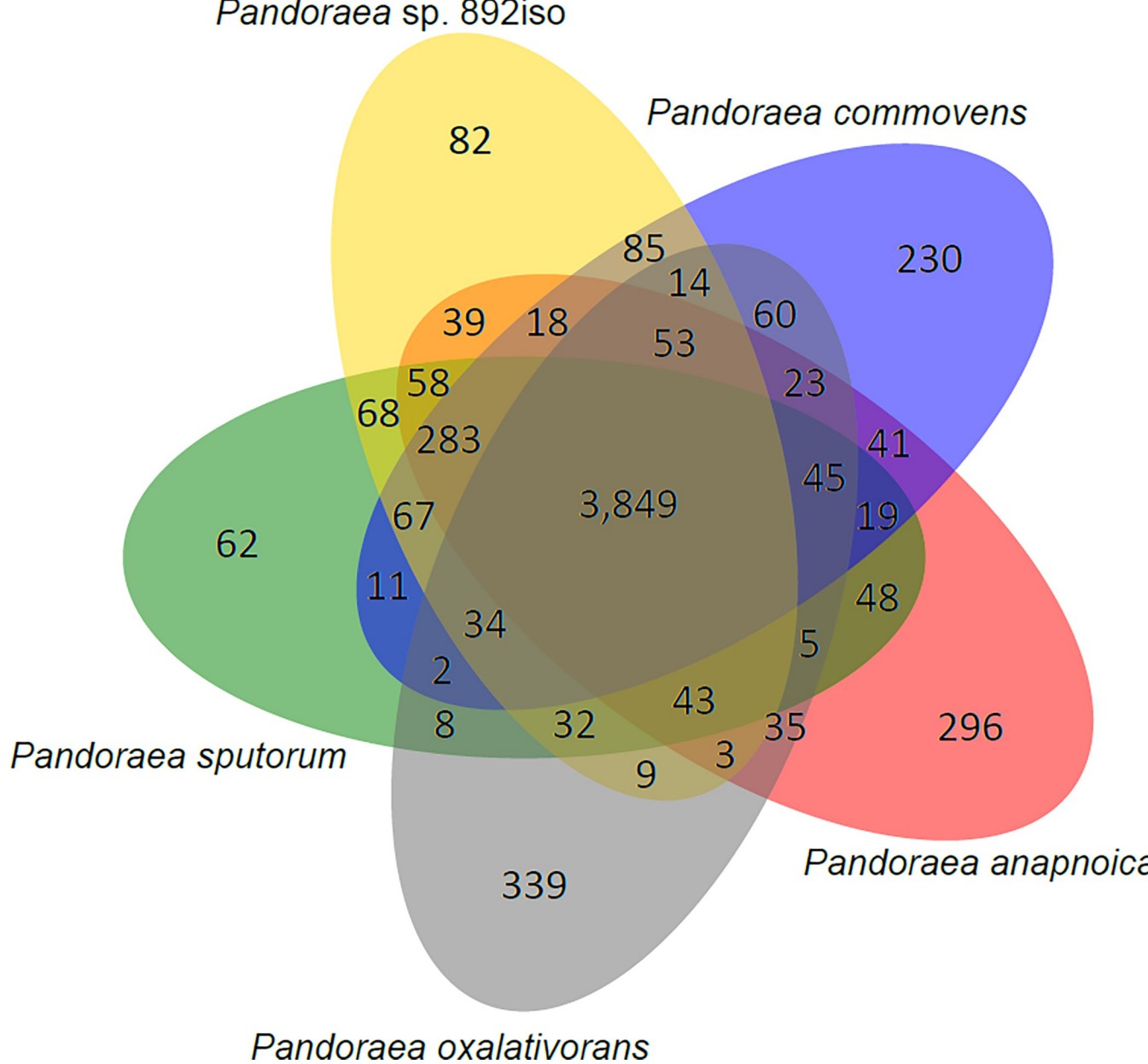

**Fig 2. Venn diagram of genes common to *Pandoraea* sp. 892iso and the four other *Pandoraea* types.**

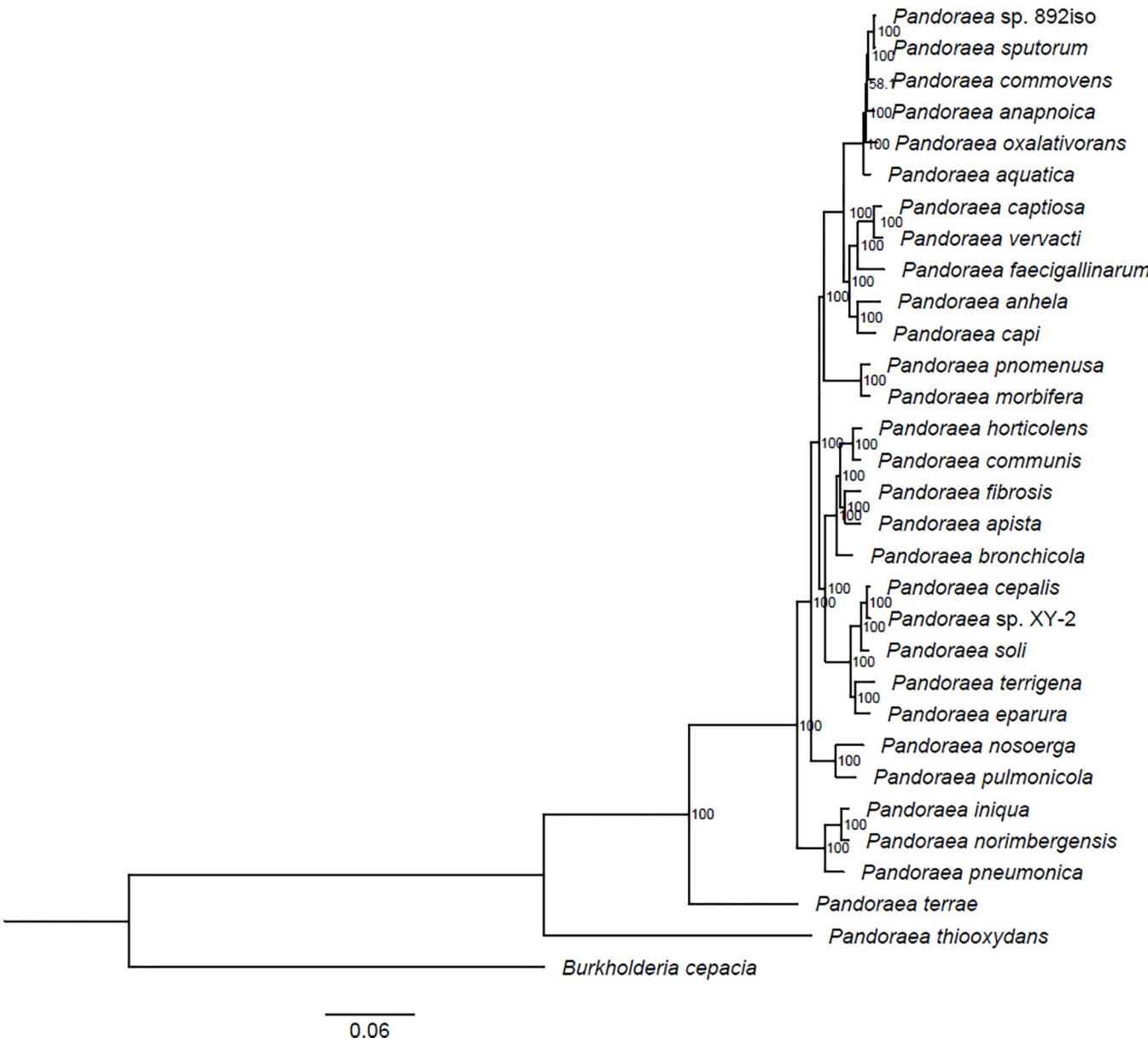

**Fig 3. Phylogenetic analyses of the evolutionary relationships between *Pandoraea* sp. 892iso and *Pandoraea* types.** A neighbor-joining phylogenetic tree constructed based on single-copy genes common to these nine bacterial genomes. The neighbor-joining method was used in MEGA6, where a bootstrap test (1,000 replicates) is shown next to the branches.

Meanwhile, eight specific gene families, including 21 genes, were clustered, 17 genes were hypothetical proteins, and the other four are shown in Table 2.

The global genome clustering and alignment of *Pandoraea* types were complicated by Mummer. The results showed the best gene co-linearity among these *Pandoraea* types and that rearrangement was almost absent, except for *P. thiooxydans* and *P. sputorum*, which were phylogenetically closest to *Pandoraea* sp. 892iso (S1 Fig). It was speculated that the small external selection pressure of the *Pandoraea* group and the genome evolution occurred in a similar way. More attention should be given to *Pandoraea* sp. 892iso and its proximal *P. sputorum*, both of which rearranged compared to other *Pandoraea* types. Rearrangements existed in the

**Table 2. Details of the four respective genes of *Pandoraea* sp. 892iso.**

| Gene | Position | direction | Detail |
|------|----------|-----------|--------|
| fig\|93222.8.peg.1 | C163_3_104 | - | DNA-cytosine methyltransferase |
| fig\|93222.8.peg.10 | C237_1_126 | - | DNA-cytosine methyltransferase |
| fig\|93222.8.peg.14 | C273_3_104 | - | DNA-cytosine methyltransferase |
| fig\|93222.8.peg.2519 | scaffold3_1641978_1640653 | - | DNA-cytosine methyltransferase |
| fig\|93222.8.peg.4329 | scaffold5_320715_322067 | + | DNA-cytosine methyltransferase |
| fig\|93222.8.peg.4363 | scaffold5_361619_362311 | + | Transcriptional regulator, GntR family |
| fig\|93222.8.peg.4364 | scaffold5_363002_362316 | - | Transcriptional regulator, GntR family |

five largest scaffold alignments, especially in scaffolds 3, 4, and 5, as shown in S2 Fig. A special unique insertion sequence in scaffold3_1802763_1803544 of *Pandoraea* sp. 892iso contains the gene fig\|93222.8.peg.2650 with the function of ubiquitin in the NR database, which may be related to the function of covalent attachment to other cellular proteins associated with stability changing, localization, and activity of the target protein [35]. The ubiquitin gene in *Pandoraea* sp. 892iso was found to be different from that in human, mouse, zebrafish, rice, Arabidopsis, yeast, or other model organisms by phylogenetic analysis (Fig 4).

**ANI and AAI.** ANI (average nucleotide identity), as the new method for bacterial species definition, provides several benefits, avoids misplacement based on phenotypic similarities or chemical characteristics, provides a scalable and uniform approach that works for both culturable and nonculturable species, is faster and cheaper than traditional taxonomic methods, and, most importantly, falls in line with Darwin's vision of classification [30]. AAI (average amino acid identity), a method that compares all conserved protein-coding genes present in a given set of genomes, clusters types into groups that share more than 95% AAI [36]. ANI and AAI characteristics have been used to evaluate the accuracy of these genotypic methods in the identification of *Pandoraea* species. Given the availability of whole genome sequence data and *Pandoraea* sp. 892iso nucleotide and amino acid data as query, Blastn by CDS sequence coverage was ≥ 50% and tblastn by protein coverage was ≥ 70%. We performed sequence-based genotypic microbial identification analysis using the RefSeq database by genome comparison between *Pandoraea* sp. 892iso and *Pandoraea sputorum* and generated an ANI value of 98.81% and an AAI value of 91.18%; genome comparison with other in-house sequenced *Pandoraea*

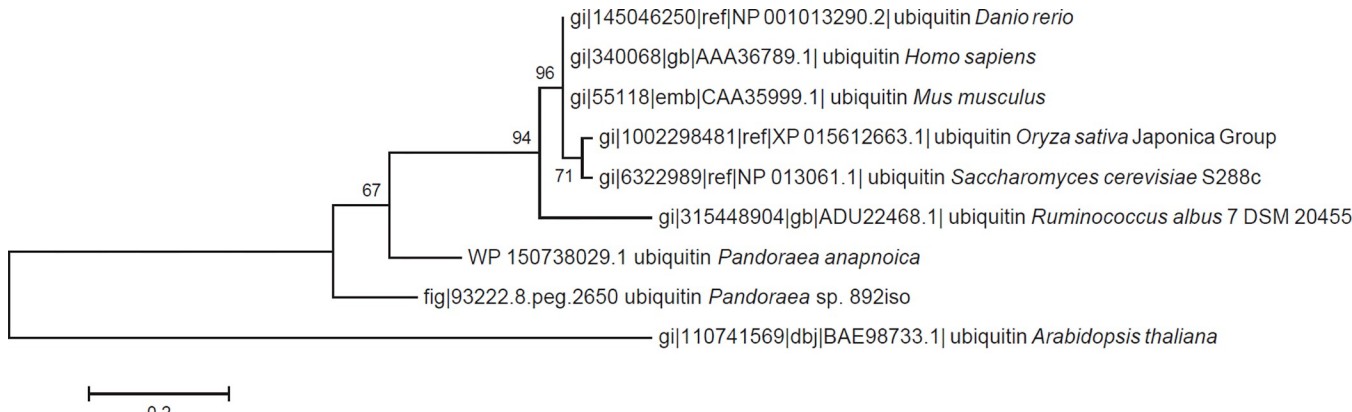

**Fig 4. Phylogenetic analyses of evolutionary relationships of ubiquitin genes among *Pandoraea* sp. 892iso and *Pandoraea* types.** A neighbor-joining phylogenetic tree constructed based on single-copy genes common to these nine bacterial genomes. The neighbor-joining method was used in MEGA6, where a bootstrap test (1,000 replicates) is shown next to the branches.

**Table 3. Average nucleotide identity (ANI) and average amino acid identity (AAI) analyses.** Genome comparisons of *Pandoraea* sp. 892iso and other *Pandoraea*-type species.

| Species | ID | ANI | | AAI | |
|---|---|---|---|---|---|
| | | value | percent | value | percent |
| *Pandoraea anapnoica* | GCF_902459765.1 | 94.10 | 83.55 | 93.57 | 89.88 |
| *Pandoraea anhela* | GCF_902459655.1 | 87.56 | 61.88 | 85.97 | 84.01 |
| *Pandoraea apista* | GCF_001465595.2 | 86.43 | 54.74 | 84.98 | 83.38 |
| *Pandoraea aquatica* | GCF_902459565.1 | 92.99 | 83.01 | 93.06 | 89.25 |
| *Pandoraea bronchicola* | GCF_902459805.1 | 86.63 | 55.10 | 84.51 | 81.14 |
| *Pandoraea capi* | GCF_902459735.1 | 87.77 | 67.77 | 88.02 | 87.59 |
| *Pandoraea captiosa* | GCF_902459775.1 | 87.24 | 61.15 | 86.49 | 85.41 |
| *Pandoraea cepalis* | GCF_902459625.1 | 86.48 | 48.39 | 82.57 | 77.98 |
| *Pandoraea commovens* | GCF_902459615.1 | 94.51 | 85.62 | 94.43 | 90.55 |
| *Pandoraea communis* | GCF_902459745.1 | 86.60 | 53.85 | 83.93 | 82.50 |
| *Pandoraea eparura* | GCF_902459725.1 | 86.58 | 48.85 | 81.94 | 77.51 |
| *Pandoraea faecigallinarum* | GCF_001029105.3 | 87.43 | 60.07 | 85.94 | 83.10 |
| *Pandoraea fibrosis* | GCF_000807775.2 | 86.50 | 57.54 | 85.87 | 83.29 |
| *Pandoraea horticolens* | GCF_902459555.1 | 86.53 | 53.29 | 83.58 | 82.45 |
| *Pandoraea iniqua* | GCF_902459685.1 | 85.58 | 54.26 | 83.99 | 86.77 |
| *Pandoraea morbifera* | GCF_902459575.1 | 86.21 | 51.70 | 83.57 | 82.34 |
| *Pandoraea norimbergensis* | GCF_001465545.3 | 85.45 | 53.68 | 83.84 | 86.68 |
| *Pandoraea nosoerga* | GCF_902459585.1 | 86.24 | 50.42 | 82.72 | 78.67 |
| *Pandoraea oxalativorans* | GCF_000972785.3 | 93.57 | 77.73 | 90.99 | 86.06 |
| *Pandoraea pneumonica* | GCF_902459645.1 | 85.60 | 52.62 | 83.58 | 85.52 |
| *Pandoraea pnomenusa* | GCF_000504585.2 | 86.27 | 53.21 | 83.90 | 82.50 |
| *Pandoraea pulmonicola* | GCF_000815105.2 | 86.28 | 53.72 | 83.78 | 82.39 |
| *Pandoraea soli* | GCF_902459595.1 | 86.52 | 48.07 | 82.37 | 77.75 |
| *Pandoraea* sp. XY-2 | GCF_004193915.1 | 86.49 | 48.39 | 80.31 | 74.75 |
| *Pandoraea sputorum* | GCF_900187205.1 | 99.29 | 88.49 | 97.03 | 90.91 |
| *Pandoraea terrae* | GCF_902459695.1 | 82.67 | 25.17 | 72.62 | 73.04 |
| *Pandoraea terrigena* | GCF_902459705.1 | 86.36 | 48.80 | 82.03 | 79.21 |
| *Pandoraea thiooxydans* | GCF_001017775.3 | 80.09 | 10.79 | 67.18 | 62.49 |
| *Pandoraea vervacti* | GCF_000934605.2 | 87.33 | 60.28 | 86.32 | 83.73 |

species provided an ANI value of less than 93.34% and an AAI value of 84.90% (Table 3). Based on previous results using the ANI value for species definition, ANI and AAI values of ≥ 95% corresponded to the traditional 70% DNA-DNA. Using the ANI and AAI values of *Pandoraea* sp. 892iso, it can be unequivocally stated that *Pandoraea* sp. 892iso is phylogenetically close to *P. sputorum*.

**_rpoB_ similarity and MLSA phylogenetic analysis.** The *rpoB* gene, encoding the β-subunit of RNA polymerase, has emerged as a core gene candidate for phylogenetic analyses and identification of bacteria; it is a single-copy gene, belongs to the common set of genes, and is long enough to contain phylogenetically useful information for some bacterial declination [37–40]. Multilocus sequence analysis (MLSA) is a currently widely used method for prokaryotic taxonomy, which utilizes internal fragments of several protein-coding genes. It was introduced by Gevers et al. and is increasingly being applied to obtain higher resolution power among species within a genus [39, 41]. As a typing technique for type characterization that shows variation in multiple housekeeping genes, a concatenation of five housekeeping genes, *shikimate dehydrogenase* (*aroE*), *guanylate kinase* (*gmk*), *phosphate acetyltransferase* (*pta*), *triosephosphate*

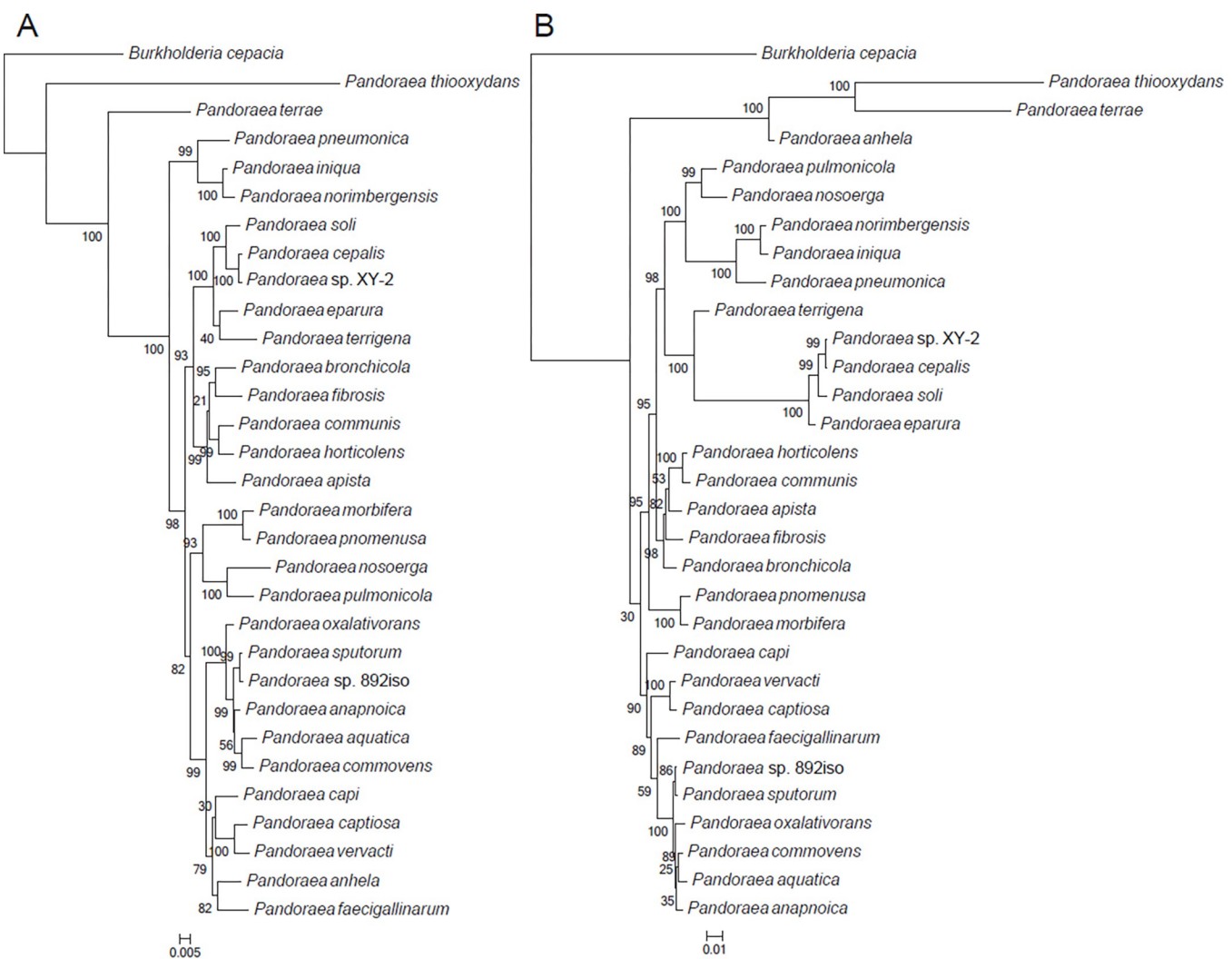

**Fig 5. Phylogenetic tree highlighting the position of *Pandoraea* sp. 892iso relative to the other *Pandoraea* species.** The tree was aligned with the characteristics of the *rpoB* gene (a) and MLSA (b) under the maximum likelihood (ML) criterion.

isomerase (*tpi*), and *acetyl coenzyme A acetyltransferase* (*yqiL*), was recommended for our bacterial delineation, as well as for clarifying the taxonomic situation within the *Pandoraea* family [39, 41]. The phylogenetic tree topologies of *Pandoraea* sp. 892iso and other *Pandoraea* spp. by *rpoB* similarity (Fig 5A) and MLSA analysis (Fig 5B) revealed *Pandoraea* sp. 892iso to have the closest phylogenetic relationship with *Pandoraea sputorum* strain DSM21091.

**Genome-to-genome distance calculator.** *In silico* genome-to-genome comparison to obtain an estimate of the overall similarity between the genomes of two types has enabled the taxonomist to perform genome-based species delineation and genome-based subspecies delineation. These distance functions can also cope with heavily reduced genomes and repetitive sequence regions. The Genome-to-Genome Distance Calculator (GGDC) calculates the distances by comparing genomes to obtain HSPs (high-scoring segment pairs) and interfering distances from a set of formulas: 1) HSP length/total length; 2) identities/HSP length; and 3) identities/total length [42]. An estimated GGDC of the overall similarity between *Pandoraea* sp. 892iso and other *Pandoraea* species is shown in Table 4. In probability DDG ≥70% index

**Table 4. Pairwise comparison of *Pandorarae* sp. 892iso and *Pandoraea* species using the GGDC.**

| Query | Reference | ID | HSP length/total length | | | | identities/HSP length | | | | identities/total length | | | | G+C difference |
|---|---|---|---|---|---|---|---|---|---|---|---|---|---|---|---|
| | | | Distance | DDH estimate (GLM-based) | Prob. DDH>70% | Prob. DDH>79% | Distance | DDH estimate (GLM-based) | Prob. DDH>70% | Prob. DDH>79% | Distance | DDH estimate (GLM-based) | Prob. DDH>70% | Prob. DDH>79% | |
| *Pandoraea* sp. 892iso | *Pandoraea anapnoica* | GCF_902459765.1 | 78.3 | [74.3–81.8%] | 0.1411 | 89.13 | 52.1 | [49.5–54.8%] | 0.0669 | 25.38 | 74.9 | [71.4–78.1%] | 0.1985 | 88.31 | 0.26 |
| *Pandoraea* sp. 892iso | *Pandoraea anhela* | GCF_902459655.1 | 48.1 | [44.7–51.5%] | 0.3215 | 8.92 | 29 | [26.6–31.5%] | 0.1476 | 0.07 | 42.4 | [39.4–45.4%] | 0.4217 | 0.31 | 0.69 |
| *Pandoraea* sp. 892iso | *Pandoraea apista* | GCF_001465595.2 | 48 | [44.6–51.4%] | 0.322 | 8.82 | 26.6 | [24.3–29.1%] | 0.1626 | 0.02 | 41.3 | [38.3–44.3%] | 0.4323 | 0.21 | 0.03 |
| *Pandoraea* sp. 892iso | *Pandoraea aquatica* | GCF_902459565.1 | 80.7 | [76.8–84.1%] | 0.1286 | 91.75 | 46.5 | [43.9–49.1%] | 0.0803 | 10.85 | 74.9 | [71.4–78.1%] | 0.1985 | 88.3 | 0.23 |
| *Pandoraea* sp. 892iso | *Pandoraea bronchicola* | GCF_902459805.1 | 48.6 | [45.2–52%] | 0.3173 | 9.79 | 27 | [24.6–29.5%] | 0.1603 | 0.03 | 41.8 | [38.9–44.9%] | 0.4268 | 0.26 | 0.31 |
| *Pandoraea* sp. 892iso | *Pandoraea capi* | GCF_902459735.1 | 61 | [57.3–64.6%] | 0.2337 | 45.76 | 29.3 | [27–31.8%] | 0.1458 | 0.08 | 51.6 | [48.5–54.7%] | 0.3455 | 4.22 | 0.78 |
| *Pandoraea* sp. 892iso | *Pandoraea captiosa* | GCF_902459775.1 | 48.7 | [45.3–52.2%] | 0.3164 | 9.99 | 28.4 | [26–30.9%] | 0.1515 | 0.05 | 42.6 | [39.6–45.6%] | 0.42 | 0.32 | 0.64 |
| *Pandoraea* sp. 892iso | *Pandoraea cepalis* | GCF_902459625.1 | 38.3 | [34.9–41.8%] | 0.4098 | 1.11 | 26.9 | [24.6–29.4%] | 0.1605 | 0.03 | 34.5 | [31.6–37.6%] | 0.5045 | 0.02 | 0.88 |
| *Pandoraea* sp. 892iso | *Pandoraea commovens* | GCF_902459615.1 | 85.3 | [81.6–88.4%] | 0.1056 | 95.14 | 54.2 | [51.5–56.9%] | 0.0625 | 32.2 | 81.4 | [78–84.3%] | 0.1615 | 96.5 | 0.03 |
| *Pandoraea* sp. 892iso | *Pandoraea communis* | GCF_902459745.1 | 43.9 | [40.5–47.4%] | 0.3559 | 4.04 | 26.8 | [24.4–29.3%] | 0.1614 | 0.02 | 38.5 | [35.6–41.6%] | 0.4599 | 0.08 | 0.09 |
| *Pandoraea* sp. 892iso | *Pandoraea eparura* | GCF_902459725.1 | 36.8 | [33.4–40.3%] | 0.426 | 0.75 | 27.5 | [25.1–30%] | 0.1571 | 0.03 | 33.6 | [30.6–36.7%] | 0.5161 | 0.01 | 1.02 |
| *Pandoraea* sp. 892iso | *Pandoraea faecigallinarum* | GCF_001029105.3 | 49.8 | [46.4–53.2%] | 0.3082 | 11.94 | 28.8 | [26.4–31.3%] | 0.1492 | 0.06 | 43.5 | [40.5–46.5%] | 0.4115 | 0.44 | 0.79 |
| *Pandoraea* sp. 892iso | *Pandoraea fibrosis* | GCF_000807775.2 | 52.1 | [48.6–55.5%] | 0.2918 | 16.86 | 26.7 | [24.4–29.2%] | 0.1618 | 0.02 | 44.1 | [41.1–47.1%] | 0.4064 | 0.52 | 0.16 |
| *Pandoraea* sp. 892iso | *Pandoraea horticolens* | GCF_902459555.1 | 41.6 | [38.3–45.1%] | 0.3767 | 2.47 | 26.8 | [24.5–29.3%] | 0.1612 | 0.02 | 36.9 | [33.9–40%] | 0.4771 | 0.04 | 0.35 |
| *Pandoraea* sp. 892iso | *Pandoraea iniqua* | GCF_902459685.1 | 39.5 | [36.2–43%] | 0.397 | 1.51 | 25.6 | [23.3–28.1%] | 0.1697 | 0.01 | 35 | [32–38%] | 0.4994 | 0.02 | 0.41 |
| *Pandoraea* sp. 892iso | *Pandoraea morbifera* | GCF_902459575.1 | 43.2 | [39.9–46.7%] | 0.3621 | 3.49 | 26.5 | [24.1–29%] | 0.1636 | 0.02 | 37.9 | [34.9–40.9%] | 0.4665 | 0.06 | 2 |
| *Pandoraea* sp. 892iso | *Pandoraea norimbergensis* | GCF_001465545.3 | 39.3 | [36–42.8%] | 0.3992 | 1.44 | 25.5 | [23.2–28%] | 0.1702 | 0.01 | 34.8 | [31.8–37.9%] | 0.5015 | 0.02 | 0.4 |
| *Pandoraea* sp. 892iso | *Pandoraea nosoerga* | GCF_902459585.1 | 40.8 | [37.4–44.2%] | 0.3849 | 2.03 | 26.8 | [24.4–29.3%] | 0.1615 | 0.02 | 36.3 | [33.3–39.3%] | 0.4842 | 0.03 | 3.47 |
| *Pandoraea* sp. 892iso | *Pandoraea oxalativorans* | GCF_000972785.3 | 62.6 | [58.9–66.2%] | 0.2242 | 51.57 | 49.4 | [46.8–52%] | 0.073 | 17.48 | 61 | [57.7–64.2%] | 0.2809 | 29.7 | 0.43 |
| *Pandoraea* sp. 892iso | *Pandoraea pneumonica* | GCF_902459645.1 | 39.9 | [36.6–43.4%] | 0.393 | 1.67 | 25.3 | [23–27.8%] | 0.1718 | 0.01 | 35.1 | [32.2–38.2%] | 0.4973 | 0.02 | 0.21 |
| *Pandoraea* sp. 892iso | *Pandoraea pnomenusa* | GCF_000504585.2 | 43.6 | [40.2–47%] | 0.3588 | 3.78 | 26.6 | [24.3–29.1%] | 0.1626 | 0.02 | 38.2 | [35.3–41.3%] | 0.463 | 0.07 | 2.23 |
| *Pandoraea* sp. 892iso | *Pandoraea pulmonicola* | GCF_000815105.2 | 41.3 | [37.9–44.8%] | 0.3797 | 2.29 | 26.6 | [24.2–29.1%] | 0.163 | 0.02 | 36.6 | [33.6–39.6%] | 0.4809 | 0.04 | 1.64 |
| *Pandoraea* sp. 892iso | *Pandoraea soli* | GCF_902459595.1 | 39.4 | [36–42.9%] | 0.3985 | 1.46 | 27 | [24.7–29.5%] | 0.16 | 0.03 | 35.4 | [32.4–38.4%] | 0.4948 | 0.02 | 0.96 |

*(Continued)*

**Table 4.** (Continued)

| Query | Reference | ID | HSP length/total length | | | | identities/HSP length | | | | identities/total length | | | | G+C difference |
|---|---|---|---|---|---|---|---|---|---|---|---|---|---|---|---|
| | | | Distance | DDH estimate (GLM-based) | Prob. DDH>70% | Prob. DDH>79% | Distance | DDH estimate (GLM-based) | Prob. DDH>70% | Prob. DDH>79% | Distance | DDH estimate (GLM-based) | Prob. DDH>70% | Prob. DDH>79% | |
| *Pandoraea sp. 892iso* | *Pandoraea sp. XY-2* | GCF_004193915.1 | 39.7 | [36.3–43.1%] | 0.3957 | 1.56 | 27 | [24.6–29.5%] | 0.1603 | 0.03 | 35.5 | [32.6–38.6%] | 0.4926 | 0.03 | 1.1 |
| *Pandoraea sp. 892iso* | *Pandoraea sputorum* | GCF_900187205.1 | 94.2 | [91.7–96%] | 0.0565 | 98.49 | 94 | [92.2–95.4%] | 0.0077 | 96.97 | 96.1 | [94.4–97.3%] | 0.0638 | 99.88 | 0.12 |
| *Pandoraea sp. 892iso* | *Pandoraea terrae* | GCF_902459695.1 | 18.2 | [15.1–21.7%] | 0.7652 | 0 | 22.6 | [20.3–25%] | 0.194 | 0 | 17.9 | [15.3–20.9%] | 0.8108 | 0 | 0.13 |
| *Pandoraea sp. 892iso* | *Pandoraea terrigena* | GCF_902459705.1 | 39 | [35.6–42.4%] | 0.403 | 1.31 | 26.7 | [24.4–29.2%] | 0.1621 | 0.02 | 34.9 | [32–38%] | 0.4998 | 0.02 | 0.82 |
| *Pandoraea sp. 892iso* | *Pandoraea thiooxydans* | GCF_001017775.3 | 14.2 | [11.4–17.6%] | 0.9145 | 0 | 20.2 | [18–22.6%] | 0.2177 | 0 | 14.4 | [12–17.2%] | 0.9331 | 0 | 0.54 |
| *Pandoraea sp. 892iso* | *Pandoraea vervacti* | GCF_000934605.2 | 50.9 | [47.5–54.4%] | 0.2999 | 14.26 | 28.4 | [26–30.9%] | 0.1512 | 0.05 | 44.1 | [41.1–47.2%] | 0.4058 | 0.53 | 0.87 |

**Table 5. The identified *ppn*I and *ppn*R genes in *Pandoraea* sp. 892iso and nine *Pandoraea* species.**

| Species | Accession number | scaffold | gene | start | end | strand |
|---|---|---|---|---|---|---|
| *Pandoraea* sp. 892iso | fig\|93222.8.peg.1246 | scaffold3 | *ppn*I | 215501 | 216286 | + |
| *Pandoraea* sp. 892iso | fig\|93222.8.peg.1247 | scaffold3 | *ppn*R | 216253 | 216966 | - |
| *Pandoraea oxalativorans* | WP_046292715.1 | NZ_CP011253.3 | *ppn*I | 4024825 | 4025493 | - |
| *Pandoraea oxalativorans* | WP_046293945.1 | NZ_CP011253.3 | *ppn*R | 4024031 | 4024732 | + |
| *Pandoraea anapnoica* | WP_150739377.1 | NZ_CABPSP010000011.1 | *ppn*I | 57228 | 57914 | - |
| *Pandoraea anapnoica* | WP_150739515.1 | NZ_CABPSP010000011.1 | *ppn*R | 56433 | 57134 | + |
| *Pandoraea pneumonica* | WP_150681584.1 | NZ_CABPSK010000004.1 | *ppn*I | 583193 | 583867 | - |
| *Pandoraea pneumonica* | WP_174988328.1 | NZ_CABPSK010000004.1 | *ppn*R | 582433 | 583146 | + |
| *Pandoraea morbifera* | WP_150566717.1 | NZ_CABPSD010000005.1 | *ppn*I | 208906 | 209694 | - |
| *Pandoraea morbifera* | WP_150566716.1 | NZ_CABPSD010000005.1 | *ppn*R | 208206 | 208919 | + |
| *Pandoraea sputorum* | WP_174555901.1 | NZ_LT906435.1 | *ppn*I | 1348270 | 1349055 | + |
| *Pandoraea sputorum* | WP_039402529.1 | NZ_LT906435.1 | *ppn*R | 1349022 | 1349723 | - |
| *Pandoraea terrae* | WP_150700195.1 | NZ_CABPRZ010000043.1 | *ppn*I | 19106 | 19732 | - |
| *Pandoraea terrae* | WP_150700194.1 | NZ_CABPRZ010000043.1 | *ppn*R | 18360 | 19076 | + |
| *Pandoraea vervacti* | WP_044456583.1 | NZ_CP010897.2 | *ppn*I | 4037152 | 4037835 | - |
| *Pandoraea vervacti* | WP_044458339.1 | NZ_CP010897.2 | *ppn*R | 4036372 | 4037073 | + |
| *Pandoraea captiosa* | WP_150627103.1 | NZ_CABPSQ010000011.1 | *ppn*I | 88267 | 88950 | - |
| *Pandoraea captiosa* | WP_150627162.1 | NZ_CABPSQ010000011.1 | *ppn*R | 87492 | 88193 | + |
| *Pandoraea pnomenusa* | WP_023871914.1 | NC_023018.2 | *ppn*I | 3778787 | 3779572 | - |
| *Pandoraea pnomenusa* | WP_080685145.1 | NC_023018.2 | *ppn*R | 3778087 | 3778800 | + |
| *Pandoraea commovens* | WP_174985011.1 | NZ_CABPSA010000008.1 | *ppn*I | 204518 | 205333 | + |
| *Pandoraea commovens* | WP_150666021.1 | NZ_CABPSA010000008.1 | *ppn*R | 205300 | 206013 | - |
| *Burkholderia cepacia* | WP_042976961.1 | NZ_CP045236.1 | *ppn*I | 471746 | 472354 | - |
| *Burkholderia cepacia* | WP_021162347.1 | NZ_CP045236.1 | *ppn*R | 473082 | 473801 | + |
| *Pandoraea faecigallinarum* | WP_167362711.1 | NZ_CP011807.3 | *ppn*I | 3690884 | 3691549 | - |
| *Pandoraea faecigallinarum* | WP_053059408.1 | NZ_CP011807.3 | *ppn*R | 3690044 | 3690820 | + |
| *Pandoraea capi* | WP_150721274.1 | NZ_CABPRV010000004.1 | *ppn*I | 224554 | 225237 | - |
| *Pandoraea capi* | WP_150721396.1 | NZ_CABPRV010000004.1 | *ppn*R | 223772 | 224473 | + |
| *Pandoraea norimbergensis* | WP_157125706.1 | NZ_CP013480.3 | *ppn*I | 1418662 | 1419441 | + |
| *Pandoraea norimbergensis* | WP_064675185.1 | NZ_CP013480.3 | *ppn*R | 1419408 | 1420109 | - |

analysis, the pairwise comparison of the genome with *P. sputorum* was found to be 98.49%, 96.97%, and 99.88% for the HSP length/total length, identities/HSP length, and identities/total length ratios, respectively. Thus, the close relationship of *Pandoraea* sp. 892iso and *P. sputorum* was verified.

## Some special genes among *Pandoraea* sp

**Quorum sensing (QS).** The most studied QS molecule is N-acyl homoserine lactone (AHL), which is secreted by gram-negative proteobacteria. AHLs are secreted by *Lux*I homologs until a threshold concentration of AHL is attained before they bind to *Lux*R homologs and subsequently activate a cascade of QS-regulated gene expression [43]. The predicted putative AHL synthase (*ppn*I) and AHL receptor protein (*ppn*R) in *Pandoraea* sp. 892iso and the nine *Pandoraea* species are shown in Table 5. The phylogenetic trees of putative AHL synthase (*ppn*I) and AHL receptor protein (*ppn*R) are shown in Fig 6.

**Intrinsic carbapenem-hydrolyzing oxacillinases.** Oxacillinases are serine *β*-lactamases of molecular class D. Many bacterial species could produce *OXA*-type enzymes, some of them

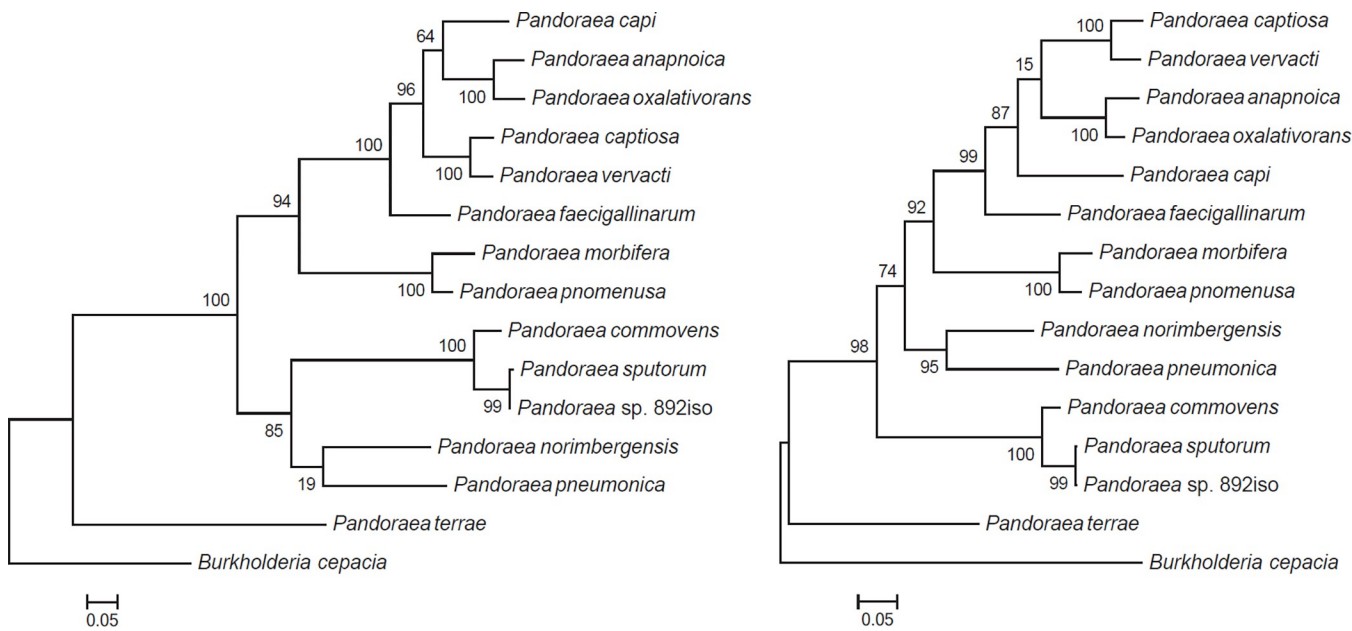

**Fig 6. Phylogenetic tree of *ppn*I and *ppn*R.**

**Table 6. The identified genes with the function of *OXA-159* in *Pandoraea* sp. 892iso and nine *Pandoraea* species.**

| Species | Accession number |
| --- | --- |
| *Pandoraea* sp. 892iso | fig\|93222.8.peg.176 |
| *Pandoraea oxalativorans* | WP_052653498.1 |
| *Pandoraea nosoerga* | WP_150556387.1 |
| *Pandoraea morbifera* | WP_150567617.1 |
| *Pandoraea sputorum* | WP_063861062.1 |
| *Pandoraea communis* | WP_150690981.1 |
| *Pandoraea fibrosis* | WP_052240481.1 |
| *Pandoraea pnomenusa* | WP_023872076.1 |
| *Burkholderia cepacia* | WP_153490194.1 |
| *Pandoraea faecigallinarum* | WP_053059421.1 |
| *Pandoraea capi* | WP_150719552.1 |
| *Pandoraea norimbergensis* | WP_058375744.1 |
| *Pandoraea anapnoica* | WP_150740206.1 |
| *Pandoraea bronchicola* | WP_150559740.1 |
| *Pandoraea iniqua* | WP_150791439.1 |
| *Pandoraea apista* | WP_048627819.1 |
| *Pandoraea pneumonica* | WP_150680540.1 |
| *Pandoraea cepalis* | WP_150607462.1 |
| *Pandoraea* sp. XY-2 | WP_130026801.1 |
| *Pandoraea pulmonicola* | WP_052266736.1 |
| *Pandoraea soli* | WP_150552526.1 |
| *Pandoraea vervacti* | WP_063389849.1 |
| *Pandoraea aquatica* | WP_150576315.1 |
| *Pandoraea captiosa* | WP_150626879.1 |
| *Pandoraea commovens* | WP_150664304.1 |
| *Pandoraea horticolens* | WP_150619975.1 |
| *Pandoraea anhela* | WP_150669648.1 |

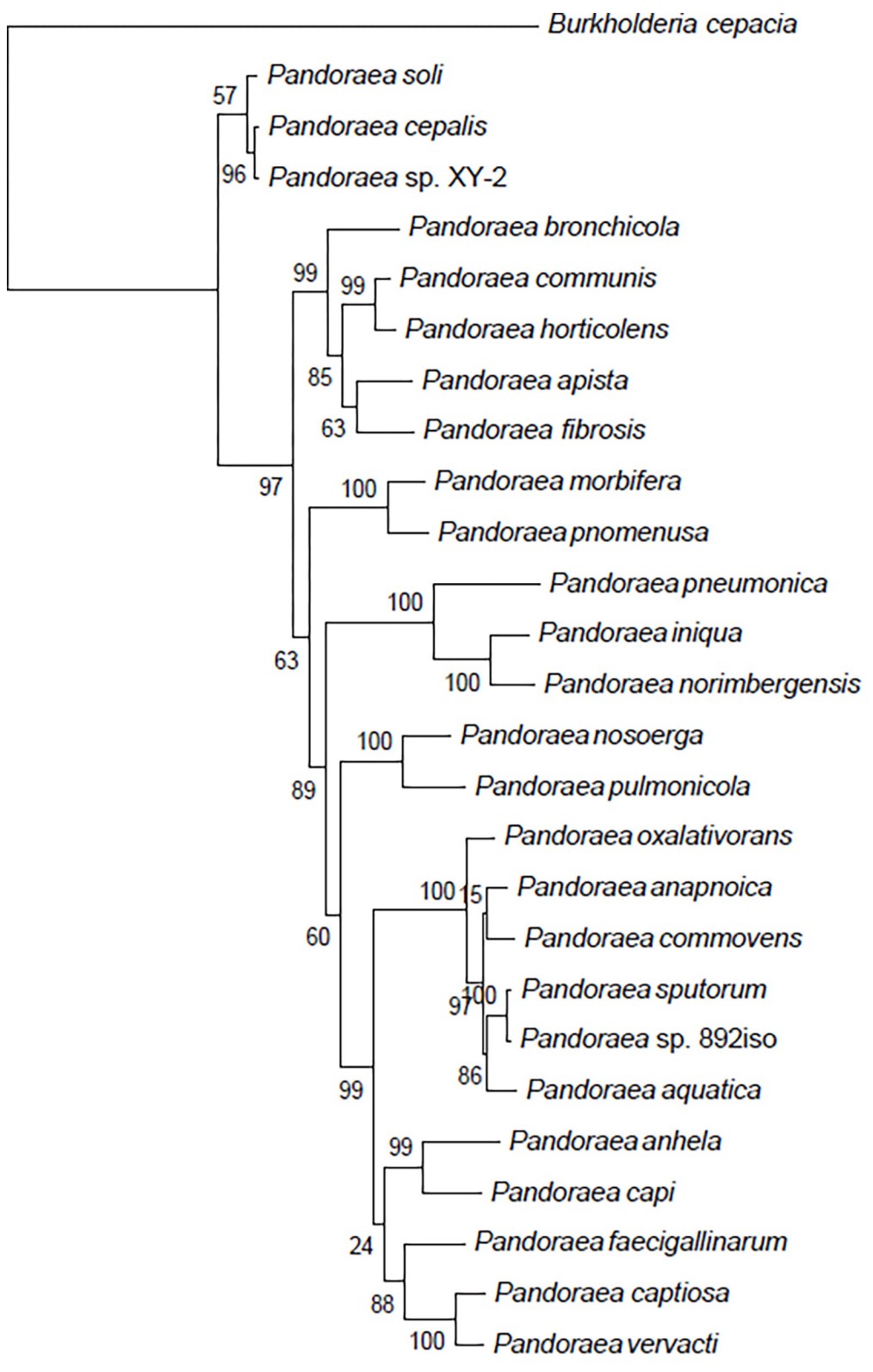

**Fig 7. Phylogenetic tree of *OXA-159* genes.** The neighbor-joining method was used in MEGA7, where a bootstrap test (1,000 replicates) is shown next to the branch.

with carbapenem-hydrolyzing activity. The nine *Pandoraea*-derived oxacillinase genes, named *OXA-159*, encode 292 amino acids and were found to be new oxacillinase variants [44]. The predicted genes with the function of *OXA-159* in *Pandoraea* sp. 892iso and the nine *Pandoraea* species are shown in Table 6. The phylogenetic trees of genes with the putative function of *OXA-159* are shown in Fig 7.

## Conclusions

We sequenced *Pandoraea* sp. 892iso from the genome of a *Phytophthora rubi* strain (numbered 109892) and combined the data with existing genomic data for other *Pandoraea* species. Next, we conducted a comparative genomic analysis of the genome structure, evolutionary relationships, and pathogenic characteristics of *Pandoraea* species. Our results identified *Pandoraea* sp. 892iso as *Pandoraea sputorum* at both the genome and gene levels. At the genome level, we carried out phylogenetic analysis of single-copy, gene co-linearity, ANI and AAI indices, *rpoB* similarity, MLSA phylogenetic analysis, and genome-to-genome distance calculator calculations to identify the relationship between *Pandoraea* sp. 892iso and *P. sputorum*. At the gene level, the quorum sensing genes *ppn*I and *ppn*R and the *OXA-159* gene were analyzed. It is speculated that *Pandoraea* sp. 892iso is the endosymbiont of the *Phytophthora rubi* strain.

## Supporting information

**S1 Fig. Diagram of linear genomic organization among *Pandoraea* types.**
(DOC)

**S2 Fig. Diagram of linear genomic organization between *Pandoraea* sp. 892iso and *Pandoraea sputorum*.** Scaffold1, scaffold2, scaffold3, scaffold4, and scaffold5 were the five largest sequences.
(DOC)

**S1 Table. The list of single copy gene in the genome of *Pandoraea* sp. 892iso.**
(XLSX)

## Author Contributions

**Data curation:** Rui-Fang Gao, Ying Wang.

**Formal analysis:** Rui-Fang Gao, Zhi-Wen Wang.

**Funding acquisition:** Ying Wang, Gui-Ming Zhang.

**Investigation:** Rui-Fang Gao.

**Methodology:** Rui-Fang Gao, Zhi-Wen Wang.

**Project administration:** Rui-Fang Gao.

**Resources:** Rui-Fang Gao, Ying Wang.

**Software:** Zhi-Wen Wang.

**Validation:** Rui-Fang Gao.

**Writing – original draft:** Rui-Fang Gao.

Writing – review & editing: Rui-Fang Gao.

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
