## [Decision Letter · Decision Letter 0]

13 Dec 2021

PONE-D-21-31328Genome insights from the identification of a Pandoraea sputorum strainPLOS ONE

Dear Dr. Gao,

Thank you for submitting your manuscript to PLOS ONE. After careful consideration, we feel that it has merit but does not fully meet PLOS ONE’s publication criteria as it currently stands. Therefore, we invite you to submit a revised version of the manuscript that addresses the points raised during the review process. Pleased, take into consideration all the comments of the reviewer. The comparison of your strain with the other bacteria has to be made with all relevant species, taking into account the updated information. Many additional species have been added to this genus in the last few years. Do not forget that you need to give all accession numbers of the sequences used.

Please, submit your revised manuscript by Jan 24 2022 11:59PM. If you will need more time than this to complete your revisions, please reply to this message or contact the journal office at plosone@plos.org. Please include the following items when submitting your revised manuscript:A rebuttal letter that responds to each point raised by the academic editor and reviewer(s). You should upload this letter as a separate file labeled 'Response to Reviewers'.A marked-up copy of your manuscript that highlights changes made to the original version. You should upload this as a separate file labeled 'Revised Manuscript with Track Changes'.An unmarked version of your revised paper without tracked changes. You should upload this as a separate file labeled 'Manuscript'

We look forward to receiving your revised manuscript.

Kind regards,

Paula V Morais, Ph.D

Academic Editor

PLOS ONE

Journal Requirements:

This work was supported by National Key R&D Programme of China (No. 2016YFF0203204) and National Key Technology Research and Development Programme of China (No. 2012BAK11B06) for G.-M. Z. Scientific research project of General Administration of Customs. P. R. China (2021HK171) for Y. W.

5. Please ensure that you refer to Figure 1 in your text as, if accepted, production will need this reference to link the reader to the figure.

6. We note you have included a table to which you do not refer in the text of your manuscript. Please ensure that you refer to Table 1 in your text; if accepted, production will need this reference to link the reader to the Table.

7.  Thank you for submitting the above manuscript to PLOS ONE. During our internal evaluation of the manuscript, we found significant text overlap between your submission and the following previously published works, some of which you are an author.

- https://journals.asm.org/doi/10.1128/AAC.01112-15

- https://www.cell.com/heliyon/fulltext/S2405-8440(21)00422-9?_returnURL=https%3A%2F%2Flinkinghub.elsevier.com%2Fretrieve%2Fpii%2FS2405844021004229%3Fshowall%3Dtrue

- https://pubmed.ncbi.nlm.nih.gov/15571809/

Please revise the manuscript to rephrase the duplicated text, cite your sources, and provide details as to how the current manuscript advances on previous work. Please note that further consideration is dependent on the submission of a manuscript that addresses these concerns about the overlap in text with published work.

Reviewers' comments:

Reviewer's Responses to Questions

**Comments to the Author**

1. Is the manuscript technically sound, and do the data support the conclusions?

Reviewer #1: Partly

2. Has the statistical analysis been performed appropriately and rigorously? 

Reviewer #1: N/A

3. Have the authors made all data underlying the findings in their manuscript fully available?

Reviewer #1: Yes

4. Is the manuscript presented in an intelligible fashion and written in standard English?

Reviewer #1: Yes

5. Review Comments to the Author

Reviewer #1: PONE-D-21-31328: Genome insights from the identification of a Pandoraea sputorum strain

The authors describe the relationship between a strain of Pandoraea derived from Phytophthora rubi with some previously described Pandoraea species at the genomic level and conclude that it is likely to belong to the species P. sputorum. They also compare quorum sensing and oxacillinase genes from their strain with those from other species of this genus.

Major comments: While the authors have been quite systematic in their comparison of strain Pandoraea sp. 892iso with other species at the genomic level, they have not included a large number of species which have been added to this genus in the last few years. In addition, while the analysis is thorough, a considerable portion of their manuscript focuses on the description of where this strain sits in the taxonomy of this genus and this seems to be out of proportion with respect to the final conclusion i.e. that this strain belongs to P. sputorum. It would be better if the authors focused less on the where their strain fits in this genus and more on the interesting aspects concerning their quorum sensing and oxacillinase gene comparisons and their MLSA typing approach.

Minor comments:

Please avoid using the word “strain”, unless required, or else preface it with “type” or “reference” where appropriate (for example in the legend for figure 3).

Title:

This should be more informative.

Abstract:

Line 13-would be useful to explain what Phytophthora rubi is.

Line 15: Would be useful to clarify which Pandoraea species you are referring to.

Line 21: Please make it clear which Oomycetes strain you are referring to.

Introduction:

Line 27: Use “genus” rather than “species”

Line 30: Please re-structure this sentence as it is unclear.

Line 33: Many additional species have been added to this genus in the last few years. Please see Peeters et al., (2019) Comparative Genomics of Pandoraea, a Genus Enriched in Xenobiotic Biodegradation and Metabolism.Front. Microbiol. 10:2556 and the following for additional species: Anandham et al., 2010; Sahin et al., 2011; Jeong et al., 2016). Please update your manuscript with this information.

Line 40: “Further”

Line 67: “were identified through whole genome sequencing”

Lines 68 and 75: Please use “isolate” rather than strain

Materials and Methods:

Line 90: Do you mean “calculations” rather than “calculate”?

Line 95 and 100: please clarify what you mean by “11 species”.

Results:

Line 114: Which Pandoraea sputorum isolate are you referring to? If it is Pandoraea sp. 892iso It may be better to use the name of the strain at this stage as the evidence for the WGS-based identification has not yet been presented.

Line 134: It would be helpful if the authors included some details about the types of single copy genes chosen for their generating their phylogenetic tree, with the genes listed in a table, or in supplementary information.

Lines 183-188: The decision to choose these particular house-keeping genes should be explained.

Line 207: A much shorter title to this section should be chosen.

Figures and Tables:

Figure 2: The legend for this figure needs further information about the methods used to generate this data.

Table 1: Please clarify the accession numbers and details for the P. pulmonicola, P. sputorum genomes used, and also for the other species for which this information had not been provided.

Table 2: This table needs some description in the legend, and further details in the main text to explain what it depicts.

6. PLOS authors have the option to publish the peer review history of their article (what does this mean?). If published, this will include your full peer review and any attached files.

Reviewer #1: No

---

## [Author Response · Author response to Decision Letter 0]

19 Jan 2022

Dear reviewer and editor,

Thank you very much for your help on our manuscript entitled ‘Genome insights from the identification of a novel Pandoraea sputorum isolate and its characteristics’. 

According to your suggestion, we have revised one by one, and now we resubmit again. 

We hope our revisions will meet with approval.

Best wishes,

Sincerely yours,

Ruifang Gao

---

## [Decision Letter · Decision Letter 1]

6 Apr 2022

PONE-D-21-31328R1Genome insights from the identification of a novel Pandoraea sputorum isolate and its characteristicsPLOS ONE

Dear Dr. Gao,

Thank you for submitting your manuscript to PLOS ONE. After careful consideration, we feel that it has merit but does not fully meet PLOS ONE’s publication criteria as it currently stands. Therefore, we invite you to submit a revised version of the manuscript that addresses the points raised during the review process.

Please, take into consideration all the comments raised by me and the reviewer.

We look forward to receiving your revised manuscript.

Kind regards,

Paula V Morais, Ph.D

Academic Editor

PLOS ONE

Additional Editor Comments:

1- The number of species of Pandorea is not correct. See https://lpsn.dsmz.de/genus/pandoraea

2- To consider a strain a new species we have 3 criteria and not 2. 16S rRNA gene sequence has to be considered (introduction).

3- What made you suspect that the fungus culture was contaminated? Give some detail (material and methods).

4- You still have the number of Pandorea wrong "...KP771987.1). BLAST was employed to identify homolog genes in the 11 Pandoraea species." Did you repeat the comparison with the 28 Pandorea strains?

5- Please, clarify "...per 1 Mb, which is more/fewer than most of the other sequenced bacteria...."

6- Table 1. What was the criteria to include 30 strains of Pandorea species? What was the criteria for the order they are included? I would like to see them by alphabetic order.

7- "In total, 3,456 849

orthologous genes were shared in common between Pandoraea sp. 892iso and the other four Pandoraea straintypes." Which ones?

8- "fig|93222.8.peg.2650 with the function of ubiquitin in the NR database." Do you mean predicted function?

9- Table 3 and Table 4. List the species by alphabetic order unless you could present a relevant criterium for not doing it.

10- "...AHL receptor protein (ppnR) in Pandoraea sp. 892iso and the nine Pandoraea species are shown..." Why 9 Pandorea species?

11- Table 6. I do not understand. How many genes? How namy species type strains?

12- "...and combined the data with existing genomic data for five other Pandoraea" Is it correct?

13. "It is speculated that Pandoraea sp. 892iso is the endosymbiont of the Phytophthora rubi strain." I do not see in your manuscript any result that supports this sentence. Why the isolate is not a contaminant of your culture? This comment is related with the 3 comment. If you want to include this senetence you have to explain in detail the fungus isolation and culture and the bacteria isolation story.

Reviewers' comments:

Reviewer's Responses to Questions

**Comments to the Author**

1. If the authors have adequately addressed your comments raised in a previous round of review and you feel that this manuscript is now acceptable for publication, you may indicate that here to bypass the “Comments to the Author” section, enter your conflict of interest statement in the “Confidential to Editor” section, and submit your "Accept" recommendation.

Reviewer #2: (No Response)

2. Is the manuscript technically sound, and do the data support the conclusions?

Reviewer #2: Yes

3. Has the statistical analysis been performed appropriately and rigorously? 

Reviewer #2: Yes

4. Have the authors made all data underlying the findings in their manuscript fully available?

Reviewer #2: Yes

5. Is the manuscript presented in an intelligible fashion and written in standard English?

Reviewer #2: Yes

6. Review Comments to the Author

Reviewer #2: There should be unification for the term that used after Pandoraea. Three different terms (types, sp., and previously strains) were employed throughout the manuscript. The term “species” is usually used in the literature.

7. PLOS authors have the option to publish the peer review history of their article (what does this mean?). If published, this will include your full peer review and any attached files.

Reviewer #2: No

---

## [Author Response · Author response to Decision Letter 1]

12 May 2022

Dear editor,

Thank you very much for your help on our manuscript entitled ‘Genome insights from the identification of a novel Pandoraea sputorum isolate and its characteristics’. 

A list of the single copy genes in the genome of Pandoraea sp. 892iso had been shown in Supporting Information files, which was noted in the section Comparative genomic analysis in the manuscript. 

"The phylogenetic relationship in view of these single-copy genes is shown in Fig 3 and S1 Table, ....."

The separate caption of S1 Table also has been added at the end of manuscript.

We hope our revisions will meet with approval.

Best wishes,

Sincerely yours,

Ruifang Gao

---

## [Editor Report · Decision Letter 2]

3 Jun 2022

PONE-D-21-31328R2

Genome insights from the identification of a novel Pandoraea sputorum isolate and its characteristics

PLOS ONE

Dear Dr. Gao,

Thank you for submitting your manuscript to PLOS ONE. After careful consideration, we feel that it has merit but does not fully meet PLOS ONE’s publication criteria as it currently stands. Therefore, we invite you to submit a revised version of the manuscript that addresses the points raised during the review process.

Figures 4 and 5 need to be improved so they can be published as supplementary figures. The manuscript has to be revised to be adapted and the figures numbered according.

We look forward to receiving your revised manuscript.

Kind regards,

Paula V Morais, Ph.D

Academic Editor

PLOS ONE
---

## [Author Response · Author response to Decision Letter 2]

4 Jun 2022

Dear editor,

Thank you very much for your help on our manuscript entitled ‘Genome insights from the identification of a novel Pandoraea sputorum isolate and its characteristics’. 

According to your suggestion, we have revised and now we resubmit again. 

We hope our revisions will meet with approval.

Best wishes,

Sincerely yours,

Ruifang Gao

---

## [Editor Report · Decision Letter 3]

20 Jul 2022

Genome insights from the identification of a novel Pandoraea sputorum isolate and its characteristics

PONE-D-21-31328R3

Dear Dr. Gao,

We’re pleased to inform you that your manuscript has been judged scientifically suitable for publication and will be formally accepted for publication once it meets all outstanding technical requirements.

Kind regards,

Paula V Morais, Ph.D

Academic Editor

PLOS ONE
---

## [Editor Report · Acceptance letter]

27 Jul 2022

PONE-D-21-31328R3 

Genome insights from the identification of a novel *Pandoraea sputorum* isolate and its characteristics 

Dear Dr. Gao:

I'm pleased to inform you that your manuscript has been deemed suitable for publication in PLOS ONE. Congratulations! Your manuscript is now with our production department. 

Kind regards, 

on behalf of

Professor Paula V Morais 

Academic Editor

PLOS ONE